# Predicting Histopathological Grading of Adult Gliomas Based On Preoperative Conventional Multimodal MRI Radiomics: A Machine Learning Model

**DOI:** 10.3390/brainsci13060912

**Published:** 2023-06-05

**Authors:** Peng Du, Xiao Liu, Xuefan Wu, Jiawei Chen, Aihong Cao, Daoying Geng

**Affiliations:** 1Department of Radiology, Huashan Hospital, Fudan University, Shanghai 200040, China; 2Department of Radiology, the Second Affiliated Hospital of Xuzhou Medical University, Xuzhou 221000, China; 3School of Computer and Information Technology, Beijing Jiaotong University, Beijing 100044, China; 4Department of Radiology, Shanghai Gamma Hospital, Shanghai 200040, China; 5Department of Neurosurgery, Huashan Hospital, Fudan University, Shanghai 200040, China

**Keywords:** histopathological grading, adult glioma, multimodal, MRI, radiomics, machine learning

## Abstract

Purpose: The accurate preoperative histopathological grade diagnosis of adult gliomas is of great significance for the formulation of a surgical plan and the implementation of a subsequent treatment. The aim of this study is to establish a predictive model for classifying adult gliomas into grades 2–4 based on preoperative conventional multimodal MRI radiomics. Patients and Methods: Patients with pathologically confirmed gliomas at Huashan Hospital, Fudan University, between February 2017 and July 2019 were retrospectively analyzed. Two regions of interest (ROIs), called the maximum anomaly region (ROI1) and the tumor region (ROI2), were delineated on the patients’ preoperative MRIs utilizing the tool ITK-SNAP, and Pyradiomics 3.0 was applied to execute feature extraction. Feature selection was performed utilizing a least absolute shrinkage and selection operator (LASSO) filter. Six classifiers, including Gaussian naive Bayes (GNB), random forest (RF), K-nearest neighbor (KNN), support vector machine (SVM) with a linear kernel, adaptive boosting (AB), and multilayer perceptron (MLP) were used to establish predictive models, and the predictive performance of the six classifiers was evaluated through five-fold cross-validation. The performance of the predictive models was evaluated using the AUC and other metrics. After that, the model with the best predictive performance was tested using the external data from The Cancer Imaging Archive (TCIA). Results: According to the inclusion and exclusion criteria, 240 patients with gliomas were identified for inclusion in the study, including 106 grade 2, 68 grade 3, and 66 grade 4 gliomas. A total of 150 features was selected, and the MLP classifier had the best predictive performance among the six classifiers based on T2-FLAIR (mean AUC of 0.80 ± 0.07). The SVM classifier had the best predictive performance among the six classifiers based on DWI (mean AUC of 0.84 ± 0.05); the SVM classifier had the best predictive performance among the six classifiers based on CE-T1WI (mean AUC of 0.85 ± 0.06). Among the six classifiers, based on ROI1, the MLP classifier had the best prediction performance (mean AUC of 0.78 ± 0.07); among the six classifiers, based on ROI2, the SVM classifier had the best prediction performance (mean AUC of 0.82 ± 0.07). Among the six classifiers, based on the multimodal MRI of all the ROIs, the SVM classifier had the best prediction performance (average AUC of 0.85 ± 0.04). The SVM classifier, based on the multimodal MRI of all the ROIs, achieved an AUC of 0.81 using the external data from TCIA. Conclusions: The prediction model, based on preoperative conventional multimodal MRI radiomics, established in this study can conveniently, accurately, and noninvasively classify adult gliomas into grades 2–4, providing certain assistance for the precise diagnosis and treatment of patients and optimizing their clinical management.

## 1. Introduction

In 2021, the WHO central nervous system (CNS) tumor classification criteria introduced molecular diagnosis indicators, emphasizing the role of molecular typing in the classification and grading of gliomas. It is suggested that CNS tumors should be diagnosed hierarchically, including histopathological grading and molecular typing. It is believed that this can more accurately guide neuro-oncologists in formulating plans for surgery, adjuvant radiation therapy, and chemotherapy [1,2,3,4].

According to the latest classification criteria, adult diffuse gliomas are classified into grades 2, 3, and 4, with different invasiveness and infiltration characteristics, and there are also significant differences in treatment plans and prognoses [5]. Therefore, if the accurate grade diagnosis of gliomas can be obtained noninvasively before surgery, it is of great significance for the formulation of surgical plans and for the implementation of subsequent treatment plans. Nowadays, MRI has been widely used in the diagnosis of brain tumors due to its superior soft-tissue contrast and the ability to conduct multiangle and multiparameter imaging [6]. However, the previous preoperative grade diagnosis of gliomas was mostly subjectively judged by radiologists and neurosurgeons, mainly based on the location, shape, signal, and contrast enhancement performance of the tumor, and the accuracy of the judgment results largely depended on the experience of the doctors [7]. Therefore, it is necessary to find an objective, stable, and reliable classification method. For patients with gliomas, surgery and subsequent histological and molecular assessments of the specimen are irreplaceable. A comprehensive pathological diagnosis is the gold standard for the diagnosis of gliomas, and it is also the basis for selecting the subsequent treatment methods, which are unavailable before surgery. Different grades of gliomas, such as WHO grade 2 and grade 3 gliomas, may have different treatment plans. Therefore, the accurate preoperative grade prediction of gliomas may be crucial for patients and might influence the formulation of the surgical plan and the implementation of the subsequent treatment.

With continuous development and progress in interdisciplinary research, such as research in medical imaging, computer science, and applied mathematics, radiomics-related research has gradually emerged and become a research hotspot. As a form of machine learning, radiomics can transform a large amount of medical image information into data information that can be used for mining to establish and train models, providing clinical decision support [8,9,10]. At present, radiomics has been widely used in the diagnosis and prognosis prediction of brain tumors. Li et al. [11] established an MRI radiomics approach to predict survival and tumor-infiltrating macrophages in gliomas. Yan et al. [12] utilized quantitative MRI-based radiomics for noninvasively predicting molecular subtypes and survival in glioma patients. However, the lack of the standardization of acquisition parameters and inconsistent methodologies between working groups have made validations unreliable; hence, multicenter studies involving heterogenous study populations are warranted [13]. Research based on radiomics to predict the grading of gliomas mainly focuses on distinguishing between low-grade and high-grade gliomas [14,15,16], and there are few studies on the specific grading of grades 2, 3, and 4. However, only distinguishing between high and low grades of tumors is clearly insufficient for the development of treatment plans for gliomas. In addition, the studies that achieved good prediction results [17,18,19,20] mostly utilized advanced MRI sequences, such as diffusion tensor imaging (DTI), diffusion kurtosis imaging (DKI), magnetic resonance spectroscopy (MRS), and dynamic susceptibility contrast-perfusion-weighted imaging (DSC-PWI), which may affect the generalization of prediction models and may not be conducive to clinical promotion and practical application.

In this study, we established a machine learning predictive model utilizing six classifiers based on preoperative conventional multimodal MRI radiomics to conveniently, accurately, and noninvasively obtain glioma grading (grades 2–4).

## 2. Patients and Methods

### 2.1. Patients

This study was approved by the institutional review board of Huashan Hospital, Fudan University. We retrospectively analyzed the data of pathologically confirmed glioma patients that were at Huashan Hospital, Fudan University, from February 2017 to July 2019. In total, 500 patients with diffuse gliomas (55 of grade 2, 42 of grade 3, and 403 of grade 4) from The Cancer Imaging Archive (TCIA) were used for external validation.

### 2.2. Inclusion and Exclusion Criteria

Inclusion criteria: (1) patients aged above 18 years; (2) patients with histopathological diagnosis of grade 2–4 gliomas; (3) patients with brain MRI examination performed within one week of surgery; and (4) patients with brain MRI acquired using a 3.0 T scanner (Magnetom Verio; Siemens Healthineers, Erlangen, Germany) with sequences including T1-weighted imaging (T1WI), T2-weighted imaging (T2WI), T2-fluid-attenuated inversion recovery (T2-FLAIR), diffusion-weighted imaging (DWI), and CE-T1WI.

Exclusion criteria: (1) previous history of brain tumors, (2) lesion was predominant hemorrhage, (3) artifacts on MRI images, (4) other treatment before surgery, and (5) incomplete clinical data.

### 2.3. MRI Scanning Parameters

Brain MRI scanning parameters are shown in Table 1. The scanning range of all sequences covered the entire brain. The contrast agent gadodiamide (GE Pharmaceuticals) was injected through the elbow vein using a dose of 0.1 mmol per kilogram of body weight for CE-T1WI scanning. After the injection of the contrast agent, transverse CE-T1WI scanning was started immediately, and, after the injection of the contrast agent, 20 mL of physiological saline was used to rinse.

### 2.4. Data Preprocessing

First, we executed an anonymous operation on the image information of included patients. After that, to match the region of interest (ROI), with the images of all sequences, we used the linear differential resampling method in SimpleITk software package (version 2.1.1.1, https://simpleitk.readthedocs.io/en/master/index.html accessed on 26 April 2023), resampling all images to 240 × 240 × 24 with an interval of 1 × 1 × 4 mm^3^. Then, by utilizing the tool of ANTs [21] (https://github.com/ANTsX, accessed on 26 April 2023), all sequences (T1WI, T2WI, T2-FLAIR, and DWI) were registered to CE-T1WI. After that, we used SimpleITK to normalize the image grayscale value to 0–255.

### 2.5. Image Segmentation

MRI image segmentation was based on the research of Zacharaki et al. [22] and the Multimodal Brain Tumor Image Segmentation Benchmark [23], and it was implemented by a radiologist with more than 20 years of experience in neuro-oncology radiology. The tumors were segmented on T2-FLAIR and CE-T1WI axial images using ITK-SNAP [24] (version 4.0.0, PICSL, Philadelphia, PA, USA, http://www.itksnap.org/pmwiki/pmwiki.php, accessed on 26 April 2023). All tumors were delineated into two parts, called ROI1 and ROI2. ROI1 represents the maximum anomaly region (MAR), and it was delineated on T2-FLAIR image and was represented in green; ROI2 represents the tumor area (tumor), and it was delineated on CE-T1WI image, with yellow representing the enhancement area and red representing the nonenhancement area. The maximum anomaly region represents the abnormal hyperintense signal region on the T2-FLAIR. Representative tumor ROI delineation is shown in Figure 1.

### 2.6. Feature Extraction and Selection

A total of 12,918 features were extracted using Pyradiomics [25] (version 3.0, DFCI of Harvard Medical School, Boston, MA, USA, https://pyradiomics.readthedocs.io/en/latest/features.html, accessed on 26 April 2023) for 6 combinations of 2 ROIs and 3 sequences to force the extraction of 2D features with slice-averaged features instead of 3D features. We used the least absolute shrinkage and selection operator (LASSO) filter to select the most significant features.

In addition, feature selection, model construction, and validation were proceeded utilizing the scikit-learn software package (version 1.0.2, Machine Learning in Python, Saclay, France, https://scikit-learn.org/stable/, accessed on 26 April 2023) basing on Python (version 3.10.7, Python Software Foundation, Wilmington, DE, USA, https://www.python.org, accessed on 26 April 2023).

### 2.7. Classifier Evaluation and Statistical Analysis

Prediction models were built utilizing six classifiers, including Gaussian naive Bayes (GNB), random forest (RF), K-nearest neighbor (KNN), support vector machine (SVM) with linear kernel, adaptive boosting (AB), and multilayer perceptron (MLP), and the predictive performance of the six classifiers was assessed through five-fold cross-validation. Additionally, we utilized area under curve (AUC), precision (PRE), recall (REC), and F1-score to evaluate the performance of each predictive model.

## 3. Results

### 3.1. Baseline Characteristics

According to the inclusion and exclusion criteria, 240 patients were enrolled in the study, with a median age of 50.05 ± 13.82 years, including 151 males and 89 females, 106 cases of grade 2 gliomas, 68 cases of grade 3 gliomas, and 66 cases of grade 4 gliomas. The baseline characteristics of the enrolled patients are shown in Table 2.

### 3.2. Feature Selection

In total, 150 features were filtered using the LASSO filter, including first-order features (*n* = 28), grayscale co-occurrence matrix features (*n* = 33), grayscale run length matrix features (*n* = 15), grayscale size region matrix features (*n* = 45), adjacent grayscale difference matrix features (*n* = 22), and grayscale dependency matrix features (*n* = 7). According to the sequence distribution, the number of features from the CE-T1WI, DWI, and T2-FLAIR sequences was 41, 52, and 57, respectively. According to the region distribution, the number of features from the maximum anomaly region (MAR) and the tumor region (tumor) was 78 and 72, respectively.

### 3.3. Different Sequence-Based Predictive Model Performances

#### 3.3.1. T2-FLAIR-Based Predictive Model Performance

The specific evaluation parameters are shown in Table 3, and the ROC curves of the six classifiers are shown in Figure 2. The results indicate that the MLP classifier had the best classification performance.

#### 3.3.2. DWI-Based Predictive Model Performance

The specific evaluation parameters are shown in Table 4, and the ROC curves of the six classifiers are shown in Figure 3. The results indicate that the SVM classifier had the best classification performance.

#### 3.3.3. CE-T1WI-Based Predictive Model Performance

The specific evaluation parameters are shown in Table 5, and the ROC curves of the six classifiers are shown in Figure 4. The results indicate that the SVM classifier had the best classification performance.

### 3.4. Different ROI-Based Predictive Model Performances

#### 3.4.1. ROI1-Based Predictive Model Performance

The specific evaluation parameters are shown in Table 6, and the ROC curves of the six classifiers are shown in Figure 5. The results indicate that the MLP classifier had the best classification performance.

#### 3.4.2. ROI2-Based Predictive Model Performance

The specific evaluation parameters are shown in Table 7, and the ROC curves of the six classifiers are shown in Figure 6. The results indicate that the SVM classifier had the best classification performance.

### 3.5. Predictive Model Performance Based on All Sequences and ROIs

The specific evaluation parameters are shown in Table 8, and the ROC curves of the six classifiers are shown in Figure 7. The results indicate that the SVM classifier had the best classification performance.

### 3.6. External Validation

The model with the best predictive performance (the SVM classifier based on multimodal MRI using all the ROIs) was tested using the data from The Cancer Imaging Archive (TCIA), achieving an AUC of 0.81 and an ACC of 0.80, and the confusion matrix is shown in Figure 8.

## 4. Discussion

With the deepening of the cognition of the glioma genome map, the biological process, the occurrence and development, and changes in some of the molecular genetics of gliomas show great diagnostic and prognostic value [26,27,28]. In 2021, the WHO CNS tumor classification criteria formulated a new classification system for brain tumors, proposed the concept of integrated diagnosis, and focused on promoting the application of molecular and genetic diagnosis in CNS tumor classification. In a sense, the diagnosis of gliomas is stepping into the era of precision medicine [1,2]. As the basis of integrated diagnosis, the histological grading of gliomas still occupies an indispensable position. Many molecular genetics changes have more clinical significance in specific tumor grading, while some tumors with a lower histological grade are redefined as high-level tumors because of special molecular genetics changes. Therefore, the accurate histopathological grading of gliomas is the foundation for further molecular diagnosis, and molecular diagnosis can modify and improve grade diagnosis. The effective complementarity between the two can achieve the goal of the precise diagnosis of gliomas and plays a role in the clinical diagnosis and treatment management of patients.

MRI is one of the most commonly used imaging tools for CNS diseases [29,30]. In conventional MRI, the T1WI sequence reflects the T1 relaxation difference of each tissue, focusing on the anatomical structure of the tissue; the T2WI sequence reflects the T2 relaxation difference of each tissue, focusing on abnormal changes in the tissue; the T2-Fair sequence is mainly used to distinguish the free water and the bound water in the tissue; the DWI sequence is used to evaluate the diffusion movement of water molecules in the tissue, which can indicate microstructural differences in diseased tissues; and CE-T1WI uses gadolinium as a contrast agent, which can shorten the T1 relaxation time, thereby enhancing the signal and reflecting the difference in gadolinium absorption between tumor tissue and normal tissue, and high-blood-flow tissues exhibit high signal intensity during CE-T1WI scanning due to the high content of gadolinium. Typical low- and high-grade gliomas, such as grade 2 and 4 gliomas, have easily identifiable radiology manifestations on conventional MRI. However, grade 2 and grade 3 gliomas, which are at the edge of low- and high-grade grading, sometimes have similar MRI presentations, making differential diagnosis tough, especially for junior doctors.

With the vigorous development of medical imaging and computer science, radiomics has emerged, which can extract high-throughput data information that cannot be recognized by the human eye to establish and train models, providing clinical decision support. Currently, radiomics methods have been widely applied in the clinical research of cancers [31,32,33]. Among them, studies using radiomics to predict glioma grades have mostly focused on discriminating between low-grade and high-grade gliomas. The study by Ditmer et al. [15] retrospectively included 94 patients with gliomas (14 low-grade and 80 high-grade gliomas) and utilized a texture analysis based on preoperative MRI images to evaluate tumor heterogeneity to distinguish tumor grades. It was indicated that using an average 2 mm texture scale yielded the best differential diagnostic efficacy, with an AUC of 0.90 and a sensitivity and specificity of 0.93 and 0.86, respectively. Wang et al. [16] used a radiomics nomogram based on multimodal MRI (CE-T1WI, T2WI, and ADC) to grade 85 patients with gliomas, and the consistency indices in the training and validation cohorts were 0.971 and 0.961, respectively. In addition, some studies used functional MRI sequences, such as DTI, MRS, and APTWI [17,18,19,20]. These advanced imaging sequences require more high-end scanners, longer examination times, and higher examination costs, which may affect the generalization of predictive models and are not conducive to clinical promotion or practical application.

In this study, we established a radiomics model for predicting glioma grades by effectively integrating radiomics features from conventional multimodal MRI and by utilizing six classifiers. It shows that the SVM classifier based on radiomics features from all the sequences and ROIs achieved the best diagnostic performance, with an average AUC of 0.85 ± 0.04. This highlights the potential significance and value of multimodality and fusion concepts in MRI radiomics research [34,35,36]. Additionally, the SVM classifiers based on the CE-T1WI and DWI sequences also achieved good diagnostic performance, with average AUCs of 0.85 ± 0.06 and 0.84 ± 0.05, respectively. Due to the fact that the enhancement of gliomas mainly occurs in the rapidly growing area of the tumor, which is closely related to the formation of neovascularization within the tumor and the destruction of the blood–brain barrier, the scope and morphological characteristics of tumor neovascularization are some of the main reference criteria for histopathological grading [37], and CE-T1WI radiomics features may be able to accurately reflect the histopathological grading of gliomas. However, the patients with gliomas that were enrolled in this study had grade 2–3 gliomas, some of which may not have had enhancement. The DWI sequence can reflect the microstructural features closely related to the tumor grade and invasion, such as the tumor cell density, the extracellular space, and vascular infiltration [38,39]. Therefore, the radiomics features of the DWI sequence may indicate the infiltration degree and tumor grade of some lower-grade gliomas without enhancement.

However, our study still has some inevitable limitations. Firstly, selection bias is an unavoidable inherent flaw in retrospective studies, and, therefore, prospective, multicenter, and larger-scale studies are needed in the future to further test the accuracy of the prediction model. Secondly, the histopathological grade prediction model of adult gliomas established in this study requires further validation in multiple centers before it can be applied to the clinic.

## 5. Conclusions

The prediction model based on preoperative conventional multimodal MRI radiomics established in this study can conveniently, accurately, and noninvasively classify adult gliomas into grades 2–4, providing certain assistance for the precise diagnosis and treatment of patients and optimizing their clinical management.

## Figures and Tables

**Figure 1 brainsci-13-00912-f001:**
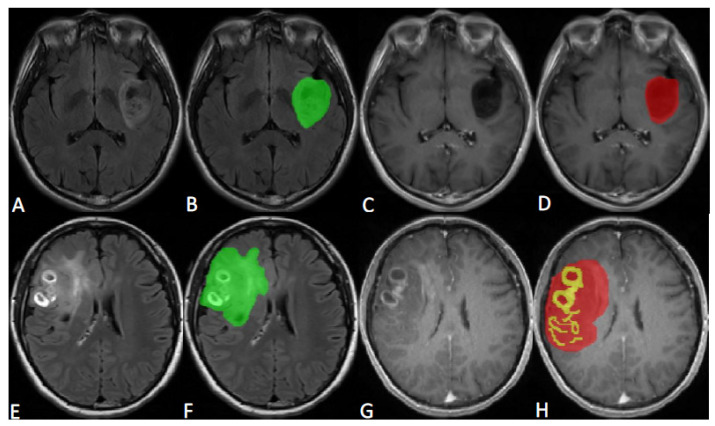
(**A**,**B**,**E**,**F**) are T2-FLAIR images, and (**C**,**D**,**G**,**H**) are CE-T1WI images. (**A**–**D**) A case of WHO grade 2 astrocytomas; B represents the maximum anomaly region (ROI1), which is delineated in green, and D represents the tumor region (ROI2), with red representing the nonenhancement region. (**E**–**H**) A case of WHO grade 4 glioblastomas; F represents the maximum anomaly region (ROI1), which is delineated in green, and H represents the tumor region (ROI2), with yellow representing the enhancement region and red representing the nonenhancement region.

**Figure 2 brainsci-13-00912-f002:**
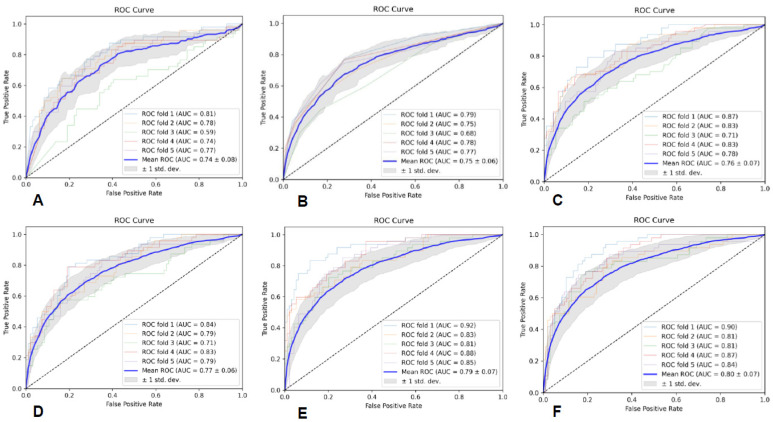
The ROC curves of the six classifiers based on T2-FLAIR; (**A**) is GNB, (**B**) is KNN, (**C**) is RF, (**D**) is AB, (**E**) is SVM, and (**F**) is MLP.

**Figure 3 brainsci-13-00912-f003:**
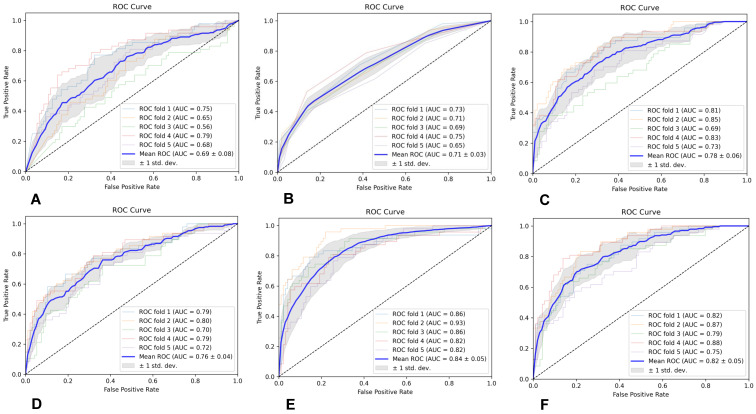
The ROC curves of the six classifiers based on DWI; (**A**) is GNB, (**B**) is KNN, (**C**) is RF, (**D**) is AB, (**E**) is SVM, and (**F**) is MLP.

**Figure 4 brainsci-13-00912-f004:**
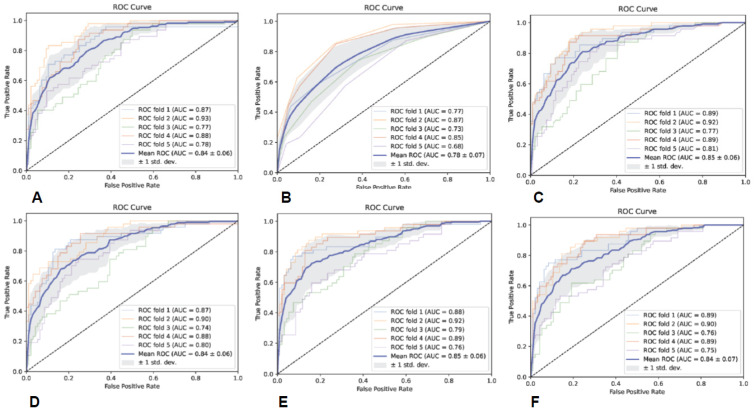
The ROC curves of the six classifiers based on CE-T1WI; (**A**) is GNB, (**B**) is KNN, (**C**) is RF, (**D**) is AB, (**E**) is SVM, and (**F**) is MLP.

**Figure 5 brainsci-13-00912-f005:**
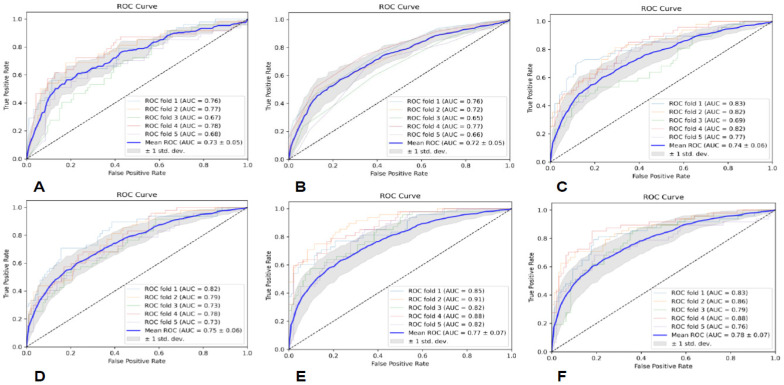
The ROC curves of the six classifiers based on ROI1; (**A**) is GNB, (**B**) is KNN, (**C**) is RF, (**D**) is AB, (**E**) is SVM, and (**F**) is MLP.

**Figure 6 brainsci-13-00912-f006:**
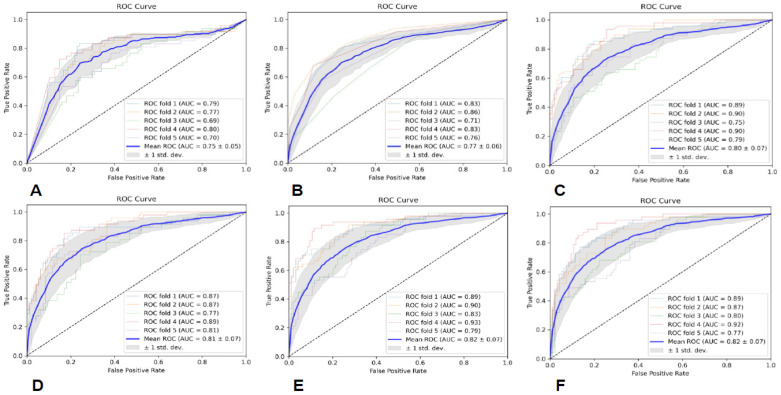
The ROC curves of the six classifiers based on ROI2; (**A**) is GNB, (**B**) is KNN, (**C**) is RF, (**D**) is AB, (**E**) is SVM, and (**F**) is MLP.

**Figure 7 brainsci-13-00912-f007:**
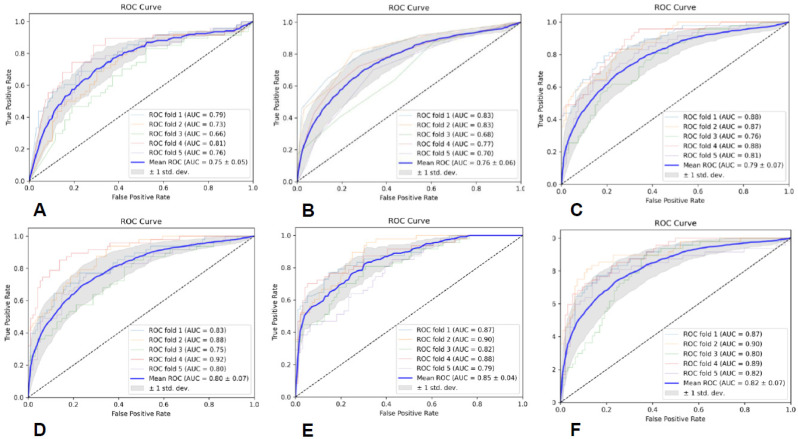
The ROC curves of the six classifiers based on all sequences and ROIs; (**A**) is GNB, (**B**) is KNN, (**C**) is RF, (**D**) is AB, (**E**) is SVM, and (**F**) is MLP.

**Figure 8 brainsci-13-00912-f008:**
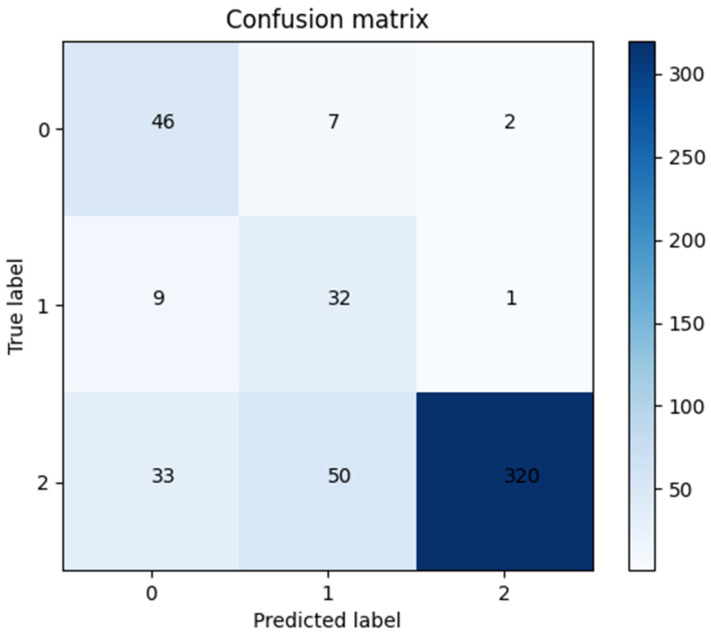
The confusion matrix of the model with the best predictive performance in external validation datasets.

**Table 1 brainsci-13-00912-t001:** Brain MRI scanning parameters.

Sequence	Parameters
T1WI SE	TR = 2000 ms; TI = 857 ms; TE = 17 ms; matrix = 256 × 168; FOV = 201 × 230 mm^2^; thickness = 5 mm; and interval = 1.5 mm
T2WI TSE	TR = 2200 ms; TE = 96 ms; matrix = 256 × 256; FOV = 260 × 260 mm^2^; thickness = 5 mm; and interval = 1.5 mm
T2-FLAIR FIR	TR = 9000 ms; TI = 2500 ms; TE = 102 ms; matrix = 256 × 190; FOV = 201 × 230 mm^2^; thickness = 5 mm; and interval = 1.5 mm
DWI EP	TR = 5000 ms; TE = 104 ms; matrix = 192 × 192; FOV = 229 × 229 mm^2^; thickness = 8 mm; and interval = 1.2 mm
CE-T1WI SE	TR = 2000 ms; TI = 857 ms; TE = 17 ms; matrix = 256 × 168; FOV = 201 × 230 mm^2^; thickness = 5 mm; and interval = 1.5 mm

Abbreviations: T1WI = T1-weighted imaging; T2WI = T2-weighted imaging; T2-FLAIR = T2-fluid-attenuated inversion recovery; DWI = diffusion-weighted imaging; CE-T1WI = contrast enhanced-T1-weighted imaging; SE = spin echo; TSE = turbo spin echo; FIR = fast inversion recovery; EP = echo planar; TR = repetition time; TI = inversion time; TE = echo time; and FOV = field of view.

**Table 2 brainsci-13-00912-t002:** The baseline characteristics of enrolled glioma patients.

Baseline Characteristics	Number of Cases (Percentage)
Total	240
Gender	
Male	151 (63%)
Female	89 (37%)
Age (year)	
Mean ± standard deviation	50.05 ± 13.82
≤50	112 (47%)
>50	128 (53%)
Histopathological grading	
WHO grade 2	106 (44%)
WHO grade 3	68 (28%)
WHO grade 4	66 (28%)

**Table 3 brainsci-13-00912-t003:** Predictive performance of six classifiers based on T2-FLAIR.

Classifier	Average AUC	Average PRE	Average REC	Average F_1_
GNB	0.74 ± 0.08	0.53 ± 0.17	0.64 ± 0.12	0.57 ± 0.14
KNN	0.75 ± 0.06	0.58 ± 0.18	0.51 ± 0.20	0.53 ± 0.17
RF	0.76 ± 0.07	0.71 ± 0.20	0.44 ± 0.19	0.52 ± 0.18
AB	0.77 ± 0.06	0.63 ± 0.16	0.48 ± 0.21	0.52 ± 0.19
SVM	0.79 ± 0.07	0.68 ± 0.16	0.61 ± 0.19	0.63 ± 0.17
MLP	0.80 ± 0.07	0.69 ± 0.19	0.62 ± 0.20	0.64 ± 0.18

Abbreviations: AUC = area under curve; PRE = precision; REC = recall; F_1_ = F_1_-score; GNB = Gaussian naive Bayes; KNN = K-nearest neighbor; RF = random forest; AB = adaptive boosting; SVM = support vector machine; and MLP = multilayer perceptron.

**Table 4 brainsci-13-00912-t004:** Predictive performance of six classifiers based on DWI.

Classifier	Average AUC	Average PRE	Average REC	Average F_1_
GNB	0.69 ± 0.08	0.47 ± 0.15	0.67 ± 0.11	0.53 ± 0.12
KNN	0.71 ± 0.03	0.55 ± 0.18	0.41 ± 0.17	0.46 ± 0.17
RF	0.78 ± 0.06	0.61 ± 0.26	0.36 ± 0.22	0.43 ± 0.22
AB	0.76 ± 0.04	0.60 ± 0.17	0.46 ± 0.19	0.50 ± 0.18
SVM	0.84 ± 0.05	0.70 ± 0.17	0.58 ± 0.18	0.62 ± 0.17
MLP	0.82 ± 0.05	0.65 ± 0.21	0.58 ± 0.18	0.60 ± 0.19

Abbreviations: AUC = area under curve; PRE = precision; REC = recall; F_1_ = F_1_-score; GNB = Gaussian naive Bayes; KNN = K-nearest neighbor; RF = random forest; AB = adaptive boosting; SVM = support vector machine; and MLP = multilayer perceptron.

**Table 5 brainsci-13-00912-t005:** Predictive performance of six classifiers based on CE-T1WI.

Classifier	Average AUC	Average PRE	Average REC	Average F_1_
GNB	0.84 ± 0.06	0.60 ± 0.16	0.71 ± 0.12	0.64 ± 0.13
KNN	0.78 ± 0.07	0.58 ± 0.19	0.51 ± 0.19	0.53 ± 0.19
RF	0.85 ± 0.06	0.70 ± 0.21	0.52 ± 0.19	0.57 ± 0.19
AB	0.84 ± 0.06	0.68 ± 0.19	0.55 ± 0.20	0.58 ± 0.19
SVM	0.85 ± 0.06	0.73 ± 0.17	0.62 ± 0.19	0.65 ± 0.18
MLP	0.84 ± 0.07	0.68 ± 0.16	0.64 ± 0.15	0.65 ± 0.15

Abbreviations: AUC = area under curve; PRE = precision; REC = recall; F_1_ = F_1_-score; GNB = Gaussian naive Bayes; KNN = K-nearest neighbor; RF = random forest; AB = adaptive boosting; SVM = support vector machine; and MLP = multilayer perceptron.

**Table 6 brainsci-13-00912-t006:** Predictive performance of six classifiers based on ROI1.

Classifier	Average AUC	Average PRE	Average REC	Average F_1_
GNB	0.73 ± 0.05	0.50 ± 0.16	0.61 ± 0.18	0.55 ± 0.16
KNN	0.72 ± 0.05	0.52 ± 0.19	0.41 ± 0.22	0.44 ± 0.20
RF	0.74 ± 0.06	0.53 ± 0.31	0.35 ± 0.26	0.41 ± 0.27
AB	0.75 ± 0.06	0.58 ± 0.22	0.41 ± 0.21	0.46 ± 0.21
SVM	0.77 ± 0.07	0.66 ± 0.19	0.62 ± 0.20	0.63 ± 0.19
MLP	0.78 ± 0.07	0.66 ± 0.18	0.64 ± 0.20	0.64 ± 0.18

Abbreviations: AUC = area under curve; PRE = precision; REC = recall; F_1_ = F_1_-score; GNB = Gaussian naive Bayes; KNN = K-nearest neighbor; RF = random forest; AB = adaptive boosting; SVM = support vector machine; and MLP = multilayer perceptron.

**Table 7 brainsci-13-00912-t007:** Predictive performance of six classifiers based on ROI2.

Classifier	Average AUC	Average PRE	Average REC	Average F_1_
GNB	0.75 ± 0.05	0.53 ± 0.16	0.78 ± 0.11	0.61 ± 0.12
KNN	0.77 ± 0.06	0.64 ± 0.15	0.59 ± 0.18	0.60 ± 0.16
RF	0.80 ± 0.07	0.73 ± 0.20	0.51 ± 0.18	0.58 ± 0.18
AB	0.81 ± 0.07	0.71 ± 0.19	0.54 ± 0.16	0.60 ± 0.16
SVM	0.82 ± 0.07	0.69 ± 0.18	0.65 ± 0.21	0.66 ± 0.19
MLP	0.82 ± 0.07	0.66 ± 0.16	0.62 ± 0.18	0.63 ± 0.17

Abbreviations: AUC = area under curve; PRE = precision; REC = recall; F_1_ = F_1_-score; GNB = Gaussian naive Bayes; KNN = K-nearest neighbor; RF = random forest; AB = adaptive boosting; SVM = support vector machine; and MLP = multilayer perceptron.

**Table 8 brainsci-13-00912-t008:** Predictive performance of six classifiers based on all sequences and ROIs.

Classifier	Average AUC	Average PRE	Average REC	Average F_1_
GNB	0.75 ± 0.05	0.53 ± 0.16	0.79 ± 0.10	0.61 ± 0.11
KNN	0.76 ± 0.06	0.54 ± 0.21	0.49 ± 0.23	0.50 ± 0.21
RF	0.79 ± 0.07	0.67 ± 0.24	0.45 ± 0.21	0.52 ± 0.21
AB	0.80 ± 0.07	0.71 ± 0.19	0.51 ± 0.19	0.57 ± 0.19
SVM	0.85 ± 0.04	0.73 ± 0.19	0.71 ± 0.19	0.70 ± 0.18
MLP	0.82 ± 0.07	0.69 ± 0.20	0.65 ± 0.22	0.65 ± 0.21

Abbreviations: AUC = area under curve; PRE = precision; REC = recall; F_1_ = F_1_-score; GNB = Gaussian naive Bayes; KNN = K-nearest neighbor; RF = random forest; AB = adaptive boosting; SVM = support vector machine; and MLP = multilayer perceptron.

## Data Availability

The data presented in this study are available on request from the corresponding author. The data are not publicly available due to protecting patient privacy.

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
