# Peer review of "Predicting Histopathological Grading of Adult Gliomas Based On Preoperative Conventional Multimodal MRI Radiomics: A Machine Learning Model"

_brainsci, 2023, doi:10.3390/brainsci13060912_

Round 1

Reviewer 1 Report

The ms by Peng et al. is a well-designed and performed research on a machine learning model predicting histopathology grading of adult glioma based on preoperative conventional multi-modal MRI radiomics.

The concept and method are clearly presented, data gathering is described, and conclusions are stated, supported by the results.

1. My minor remark is the number of subgroups - at least >60 patients in each malignancy group are generally sufficient, it would be better if the groups were more equal. Especially in the context of their molecular diversity (see point 3.).

2. I couldn't identify the anonymyzed list of patients (with age, gender, histopathological diagnosis, and molecular markers)?

3. My major remark is that the concept of distinguishing just tumor grades with MRI (even AI concept) is nowadays rather old fashioned. The revolution, that was done through 2021 classification (and earlier - 2016 revision) was introduction of molecular markers, particularly for gliomas, which better characterize the prognosis and predict the treatment outcome (e.g. AA grade 3 IDHwt has a similar prognosis as GBM grade 4 IDHmt, 10.1186/s40478-019-0817-0). The clinical decisions for gliomas are mainly done now based on those markers. Therefore distinguishing glioma grades (which is a part of the subjective aspect of intedrated diagnosis - histopathological description), even with AI-MRI, is a drawback. Especially when there are already data showing the possibility of distinguishing molecular subtypes with MRI (e.g. 10.1093/neuonc/noy048, 10.1093/noajnl/vdac023). The authors should rather follow that route or sufficiently comment on the diagnostic and therapeutic value of just distinguishing between tumor grades. Maybe those data exist and molecular subgroups in the analyzed cohort can be extracted?

Some small punctuation/grammatical/spelling/stylistic mistakes to be corrected - need one thorough reading.

Author Response

Point 1: My minor remark is the number of subgroups - at least >60 patients in each malignancy group are generally sufficient, it would be better if the groups were more equal. Especially in the context of their molecular diversity (see point 3.).

Response 1: Thanks for your comments. In this study, there were 106 patients of WHO grade 2, 68 patients of WHO grade 3, and 66 patients of WHO grade 4, and the enrolled patients were based on the inclusion and exclusion criteria. A proportion of patients with WHO grade 3 and 4 glioma were excluded because they had already received other treatment before surgery or the lesion was predominant hemorrhage.  

Point 2: I couldn't identify the anonymyzed list of patients (with age, gender, histopathological diagnosis, and molecular markers)?

Response 2: Thanks for your comments. This study mainly focuses on predicting grading of gliomas, and therefore histopathological diagnosis and molecular markers are not listed.   

Point 3: My major remark is that the concept of distinguishing just tumor grades with MRI (even AI concept) is nowadays rather old fashioned. The revolution, that was done through 2021 classification (and earlier - 2016 revision) was introduction of molecular markers, particularly for gliomas, which better characterize the prognosis and predict the treatment outcome (e.g. AA grade 3 IDHwt has a similar prognosis as GBM grade 4 IDHmt, 10.1186/s40478-019-0817-0). The clinical decisions for gliomas are mainly done now based on those markers. Therefore distinguishing glioma grades (which is a part of the subjective aspect of intedrated diagnosis - histopathological description), even with AI-MRI, is a drawback. Especially when there are already data showing the possibility of distinguishing molecular subtypes with MRI (e.g. 10.1093/neuonc/noy048, 10.1093/noajnl/vdac023). The authors should rather follow that route or sufficiently comment on the diagnostic and therapeutic value of just distinguishing between tumor grades. Maybe those data exist and molecular subgroups in the analyzed cohort can be extracted?

Response 3: Thanks for your comments. In 2021, WHO formulated a new classification system for brain tumors and proposed the concept of integrated diagnosis. As the basis of integrated diagnosis, histologic grading of gliomas still occupies an indispensable position. Many molecular alterations are more clinically significant only in specific tumor grading, while some tumors with lower histologic grading are redefined as high grade because they possess specific molecular alterations. Therefore, accurate histologic grading of glioma is the basis for further molecular diagnosis, and molecular diagnosis can modify and improve the grading diagnosis, and the effective complementation of the two can achieve the goal of accurate diagnosis of glioma. This study mainly focuses on predicting grading of gliomas, and accurate preoperative histologic grading may be of great significance for the formulation of surgical plan and the implementation of subsequent treatment. Besides, we have also conducted a study on the prediction of molecular markers of glioma based on preoperative MRI radiomics, which will be presented in a future paper.    

Special thanks to you for your precious comments.

Reviewer 2 Report

This manuscript introduces a glioma grading classification approach using pyradiomics on various MRI data of a selection of regional patients. Sseveral tools already established have been integrated into the introduced framework.

·       The main asset of this work are datasets used for modeling considering little novelty in technological aspects introduced. Therefore, at least an anonymized subset could be linked to a suitable repository.

·       Furthermore, the background could be extended and lacks some essential references such as to the Multimodal Brain Tumor Image Segmentation Benchmark. Radiomic glioma models are used for more than just differentiation between LGG and HGGs, also for survival prediction (L Guanzhang et al. doi: 10.1093/brain/awab340, J Yan et al. doi: 10.1038/s41698-021-00205-z), extended with explainabbility (M Eder et al. doi: 10.3390/biomedinformatics2030031), ultimately there is still the challenge of multi-centre studies (G Singh et al. doi: 10.1038/s41416-021-01387-w), therefore, again making data availability an issue, and should be considered.

·       Moreover, models could be tested with another dataset, such as from TCIA (https://www.cancerimagingarchive.net/).

·       Last, tools and methods have to be cited properly, including:
PA Yushkevich et al. 2016, doi: 10.1109/EMBC.2016.7591443
JJM Griethuysen et al. 2017, doi: 10.1158/0008-5472.CAN-17-0339
BB Avants et al. 2011, doi: 10.1007/s12021-011-9109-y

Continued success with your research!

Author Response

Point 1: The main asset of this work are datasets used for modeling considering little novelty in technological aspects introduced. Therefore, at least an anonymized subset could be linked to a suitable repository.

Response 1: Thanks for your comments. Our prediction model can be used for other independent datasets, and the datasets we use can also be made public. However, the hospital that provide current training and validation data require temporary confidentiality. Therefore, we will make the anonymized subset linked to a suitable repository in future. We will notify you as soon as possible after the anonymized subset datasets are made public. Thank you again for your suggestions.

Point 2: Furthermore, the background could be extended and lacks some essential references such as to the Multimodal Brain Tumor Image Segmentation Benchmark. Radiomic glioma models are used for more than just differentiation between LGG and HGGs, also for survival prediction (L Guanzhang et al. doi: 10.1093/brain/awab340, J Yan et al. doi: 10.1038/s41698-021-00205-z), extended with explainabbility (M Eder et al. doi: 10.3390/biomedinformatics2030031), ultimately there is still the challenge of multi-centre studies (G Singh et al. doi: 10.1038/s41416-021-01387-w), therefore, again making data availability an issue, and should be considered.

Response 2: Thanks for your comments. We have added relevant literature based on your suggestion.   

Point 3: Moreover, models could be tested with another dataset, such as from TCIA (https://www.cancerimagingarchive.net/).

Response 3: Thanks for your comments. According to your suggestion, we have tested the model with the best predictive performance by the external data from The Cancer Imaging Archive (TCIA), and the results have been shown in the text.

Point 4: Last, tools and methods have to be cited properly, including:

PA Yushkevich et al. 2016, doi: 10.1109/EMBC.2016.7591443

JJM Griethuysen et al. 2017, doi: 10.1158/0008-5472.CAN-17-0339

BB Avants et al. 2011, doi: 10.1007/s12021-011-9109-y

Response 4: Thanks for your comments. We have added relevant literature based on your suggestion.     

Special thanks to you for your precious comments.

Reviewer 3 Report

Major comment that needs to be addressed in a revision

1. I agree with the author's statement that accurate preoperative histopathology grading diagnosis of adult glioma is of great significance for the formulation of a surgical plan and the implementation of subsequent treatment. However, do the authors believe that the results shown in this study can replace or obviate surgery and subsequent histological and molecular assessments of the specimen for gliomas? As a background, this issue needs to be mentioned sufficiently to convince readers of the rationale of this study. Unless this manuscript is likely to be prose using statistical methods.

Specific comments

Abstract

2. "AUC" in Patients and Methods is to be spelled out.

Introduction

3. As for the description in the second paragraph, please refer to my major comment.

4. Again, in the second paragraph, “enhanced performance” should be “contrast enhancement performance”.

5. From this section throughout the manuscript, "Flair" is better to be replaced by "FLAIR", which is used worldwide.

Patients and Methods

6. In “Image segmentation”, please define the maximum anomaly region.

RESULTS

7. It is somewhat surprising that there were a lot of cases diagnosed with grade-2 gliomas (44%). Was there any patient basis?

8. As the authors may have noticed, the description of “Different sequences based predictive model performance” for the three sequences is similar in style. In addition, details in the text are presented in tables. Such a description is redundant and changed to a simpler manner. This also applies to the description in the following part of Results.

Discussion

9. In two sentences in the last part of the second paragraph, “high- and low-grade gliomas” is to be “low- and high-grade gliomas”.

Figures and Tables

10. In Table 1, names of scanning sequences, such as spin-echo, should be included for each sequence.

11. Table 8 lacks SVM and MLP.

Satisfactory.

Author Response

Point 1: Major comment: I agree with the author's statement that accurate preoperative histopathology grading diagnosis of adult glioma is of great significance for the formulation of a surgical plan and the implementation of subsequent treatment. However, do the authors believe that the results shown in this study can replace or obviate surgery and subsequent histological and molecular assessments of the specimen for gliomas? As a background, this issue needs to be mentioned sufficiently to convince readers of the rationale of this study. Unless this manuscript is likely to be prose using statistical methods.

Response 1: Thanks for your comments. For patients with glioma, surgery and subsequent histological and molecular assessments of the specimen is irreplaceable. A comprehensive pathological diagnosis is the gold standard for the diagnosis of glioma and also the basis for  selecting the subsequent treatment methods, which is unavailable before surgery. Different grades of gliomas, such as WHO grade 2 and grade 3, may have different treatment plans. Therefore, accurate preoperative grading prediction of gliomas may be crucial for patients, which might influence the formulation of the surgical plan and the implementation of subsequent treatment. We have added the above content to the introduction section and highlighted the text in blue.

Abstract

Point 2: "AUC" in Patients and Methods is to be spelled out.

Response 2: Thanks for your comments. We have made modifications in the text based on your suggestion.

Introduction

Point 3: As for the description in the second paragraph, please refer to my major comment.

Response 3: Thanks for your comments. We have supplemented the description in the second paragraph.

Point 4: Again, in the second paragraph, “enhanced performance” should be “contrast enhancement performance”. 

Response 4: Thanks for your comments. We have replaced “enhanced performance” with “contrast enhancement performance” according to your suggestion.  

Point 5: From this section throughout the manuscript, "Flair" is better to be replaced by "FLAIR", which is used worldwide. 

Response 5: Thanks for your comments. We have replaced “Flair” with “FLAIR” in the text.

Patients and Methods

Point 6: In“Image segmentation”, please define the maximum anomaly region.

Response 6: Thanks for your comments. We have added the definition of “maximum anomaly region” in “Image segmentation”.

Results

Point 7: It is somewhat surprising that there were a lot of cases diagnosed with grade-2 gliomas (44%). Was there any patient basis?

Response 7: Thanks for your comments. The enrolled patients in this study were based on the inclusion and exclusion criteria. A proportion of patients with WHO grade 3 and 4 glioma were excluded because they had already received other treatment before surgery or the lesion was predominant hemorrhage.

Point 8: As the authors may have noticed, the description of “Different sequences based predictive model performance” for the three sequences is similar in style. In addition, details in the text are presented in tables. Such a description is redundant and changed to a simpler manner. This also applies to the description in the following part of Results.

Response 8: Thanks for your comments. We have removed the redundant descriptions in the results section according to your suggestion.

Discussion

Point 9: In two sentences in the last part of the second paragraph, “high- and low-grade gliomas” is to be “low- and high-grade gliomas”.

Response 9: Thanks for your comments. We have replaced “high- and low-grade gliomas” with “low- and high-grade gliomas” according to your suggestion.  

Figures and Tables

Point 10: In Table 1, names of scanning sequences, such as spin-echo, should be included for each sequence.

Response 10: Thanks for your comments. We have supplemented the names of the scanning sequences according to your suggestion.   

Point 11: Table 8 lacks SVM and MLP.

Response 11: Thanks for your comments. We have added SVM and MLP to Table 8.  

Special thanks to you for your precious comments.

Round 2

Reviewer 2 Report

Thank you for responding to my comments. Still a few things could be improved! The manuscript should not be published until the data availability statement has been updated complying to the following stated by MDPI: “Data sharing policies concern the minimal dataset that supports the central findings of a published study. Generated data should be publicly available and cited in accordance with journal guidelines.” This would include an indication to entries in repositories for code and data enabling further citation possibilities!

Continued success with your research!

Reviewer 3 Report

It seems that the manuscript has been sufficiently improved according to my comments and suggestions.